# Some Slippage Issues in High-Pressure Torsion Using Cu and Ti Samples as an Example

**DOI:** 10.3390/ma16010162

**Published:** 2022-12-24

**Authors:** Dmitriy Gunderov, Rashid Asfandiyarov, Vyacheslav Titov, Sofia Gunderova, Vasily Astanin

**Affiliations:** 1Institute of Molecule and Crystal Physics of Ufa Federal Research Centre RAS, 151 Prospekt Oktyabrya Ave., 450075 Ufa, Russia; 2Department of Materials Science and Physics of Metals, Ufa University of Science and Technology, 32 Zaki Validi Str., 450076 Ufa, Russia

**Keywords:** HPT, slippage, commercially pure titanium, finite element method

## Abstract

The effect of slippage during High Pressure Torsion (HPT) of technically pure Ti and pure Cu samples was investigated. The “joint torsion of the disk halves” method was used to evaluate the effect of slippage. It was shown that slippage starts already at the early stages of HPT. With a further increase in the number of revolutions *n*, the slippage effect increases, and no torsional deformation occurs after *n* = 5. The slippage effect is explained by analyzing the surface friction forces between the sample and the anvil. However, studies via TEM and XRD have shown that the structure of Ti samples after HPT at the investigated conditions is grinded to a nanocrystalline state. A structure is formed in Ti similar to that observed after HPT by other authors. The dislocation density increases with increasing HPT degree from *n* = 5 to *n* = 10 revolutions, despite slippage. Consequently, despite slippage at HPT at *n* ≥ 5, deformation still occurs. The following assumptions are made to explain the accumulated strain in the sample at HPT. It is assumed that the planes of the upper and lower anvil during HPT are at a slight inclination relative to each other. Computer modeling using the Deform 3D software package has shown that this leads to the accumulations of significant strain during HPT.

## 1. Introduction

The high-pressure torsion method is used to refine the structure to a nanocrystalline state in metallic materials, which leads to a significant improvement in their properties [1,2,3]. Bridgman is the first to introduce the process of high-pressure torsion (HPT) [4]. HPT makes it possible to achieve the highest degree of strain among the known methods of plastic processing and, accordingly, to achieve the strongest grain refinement [1,2,3]. In the case of alloy processing, HPT can lead to nonequilibrium phase transformations, the formation of metastable phases [1,2,3], and even, in some cases, amorphization [4,5,6,7]. The effect of HPT on metals and alloys has been studied in a large number of articles (more than three thousand). Several large reviews are devoted to the topic of HPT, in particular [1,2,3].

In the HPT processing, a thin disc is placed in the central shallow hole on a lower anvil which is then raised to contact the upper anvil. While applying a high pressure (usually 5 GPa or more) the upper and lower anvils are rotated relatively to each other [1].

Generally, the accumulated shear strain *γ* at HPT can be estimated using the following equation [2]:(1)$$\gamma =\frac{2\pi Rn}{h},$$
where *n* is the number of revolutions, *R* is the radius from the center to the measuring point, and *h* is the thickness of the sample.

In the HPT process, the sample is subjected to quasi-hydrostatic compression under high pressure, which prevents the sample from fracturing despite the high degree of strain. During HPT, researchers use different applied pressure, number of revolutions, process temperatures, rotational speed, and geometry of the anvils [1,2,3]. In this case, materials of the same grades studied in the works of different authors, particularly pure metals, often have different impurity content and other nuances of the initial state. Thus, the results of different authors on the structure refinement of the same metal at HPT can differ appreciably.

In pure metals, as well as in alloys, HPT leads to the formation of a nanostructured state. A large number of works have been previously performed on the study of the effect of HPT on the structure of Ti [8,9,10,11,12,13,14]. In [14], more than 10 researches on the study of the HPT effect on the Ti structure were analyzed. It is shown that an ultrafine-grained structure with a mean grain size of about 100 nm is formed after HPT in commercial-purity (CP) titanium. This structure is characterized by large internal stresses. At the same time, the achieved minimum grain size varies greatly in different authors, which is associated with the difference in HPT conditions and the difference in the content of impurities in the initial Ti [8,9,10,11,12,13,14]. The microhardness HV increases with an increase in the number of revolutions at an early stage of HPT, and reaching a certain maximum of values, with a further increase in the number of revolutions, remains unchanged [13,14]. It should be noted that the maximum HV value depends on the applied pressure *P* [14]. Pure Ti in the initial state at room temperature has a hexagonal-close-packed (HCP) crystal structure [12]. HCP crystal structure at HPT under high pressure transforms to a phase with a simple hexagonal structure [12]. It is reported that the formation of a phase occurred in Ti even at HPT under a pressure of 3–4 GPa [13].

In [12], samples of technically pure Ti in the form of a disk with a diameter of 20 mm and an initial thickness of 0.2 mm were processed using the HPT method at a quasi-hydrostatic pressure of 6 GPa for 0.5, 1, 5, and 10 revolutions at an angular rotation speed of 1 rad/min. The HPT at *n* = 0.5 of a revolution and pressure of 6 GPa leads to the formation of a structural state with ω-phase content of up to 50–55%. The maximum volume fraction of ω-phase (85–90%) was achieved after deformation at a pressure of 6 GPa and *n* = 10 revolutions [12].

The HPT scheme assumes that the frictional force between the sample and anvil surfaces during rotation leads to a displacement of the upper part of the sample volume relative to the lower part of the sample and, accordingly, to a shear strain in the sample. However, it is known that at HPT of hard or strengthening materials, the so-called “slippage” of the anvil over the surface of the sample is possible, and the real shear strain *γ_real_* does not conform to the calculated one (according to Formula (1)), which is shown in several works, in particular [15,16,17,18,19,20,21]. Bridgeman [4] has already shown that slippage is possible in HPT. Slippage is caused by the fact that the frictional force between the sample and anvil surfaces becomes less than the yield strength of the material. Slippage becomes increasingly significant in the HPT of durable and hard materials, in particular BMG [18,19,20]. A widely used method for assessing slippage is the application of a scratch marker to the top and bottom surfaces of the disc subjected to HPT [15]. The real degree of shear strain *γ_real_* and the real displacement of the sample are estimated from the mutual displacement of the scratch markers. The obtained *γ_real_* is compared with the one calculated by the Formula (1) and, accordingly, the slippage at HPT is estimated. However, at HPT with an anvil rotation angle *θ* > 90°, the scratch markers are erased from the surface of the samples by deformation, and this does not allow estimating the deformation by their relative shear at HPT with >90° [15]

Another method to evaluate the slippage degree at HPT, the joint HPT of two halves of the disk, was proposed earlier [18]. In [18], using this method, it was shown that the real shear strain *γ_real_* in hard bulk metallic glass (BMG) (YTS > 1500 MPa) at HPT with *n* = 0.25–5 is much lower than expected by Formula (1). However, despite this, the structure of BMG after HPT significantly changed [18]. Similar results were obtained in [19]—it is shown that the real shear strain *γ_real_* in hard BMG at HPT with *n* = 0.25–5 is much lower than expected by Formula (1), but the structure of BMG after HPT significantly changes [19].

The authors of the present paper have used the joint HPT method of two halves for the last two years to evaluate slippage on several crystalline metallic materials [19,20,21,22]. It was shown that such a relatively “soft” material as copper in the early stages of HPT is significant *γ_real_*, and the slippage was not so substantially [22].

However, during HPT of such materials as steel Fe—0.1% C, Zr—1% Nb alloy, Ti_18_Zr_15_Nb alloy, the following was revealed [21,22,23]: the samples of these materials at initial HPT stages received a certain degree of shear strain, but significant slippage is observed already at initial HPT stages. Samples after HPT with *n* > 5, with further increase in *n*, do not receive a noticeable shear strain due to slippage. However, the structure of the alloys studied is refined and changed after HPT [21,22,23] as other authors observed in similar alloys after HPT.

At the moment, the method of joint HPT of two halves [18,22] looks promising to study; it may provide additional data for the analysis of the slippage process at HPT.

The purpose of this study is to analyze slippage at high-pressure torsion and to determine the degree of shear strain achieved, in particular on Cu and Ti, by increasing the number of HPT revolutions

## 2. Materials and Methods

This study used commercially pure titanium of the Russian brand VT1-0. The chemical composition is presented in Table 1.

Commercially pure copper was chosen as an auxiliary material for the research. For HPT, anvils with a working groove with a diameter of 20 mm and a depth of 0.5 mm, as well as flat anvils without a groove were used. Initial workpieces were made according to the diameter of the working groove in beats with a groove, the initial height of the workpieces was about 1 mm. The HPT was performed at room temperature (RT), with an operating pressure of 6 GPa.

The method of determining the degree of strain used in HPT (Figure 1a) [18,22] was as follows: the initial metal disk was cut into two halves; the ends of each segment were polished and varnished to avoid metal adhesion; the segments were placed on the anvil according to the Figure 1a. Then the combined segments of Cu and Ti were subjected to HPT. This approach is described in more detail in the results section. Furthermore, one-piece Ti discs were subjected to HPT according to the traditional scheme with different numbers of revolutions *n*.

Microhardness was measured using the Vickers method under a load of 1 *N* (100 *g*) for 10 s. X-ray diffraction (XRD) analysis was conducted on a Rigaku Ultima IV diffractometer. The samples were examined with CuK α-radiation (40 kV, 30 mA) and the phase composition of the alloy was determined using the Rietveld method. The fine structure of the samples was examined using a JEOL JEM-2100 transmission electron microscope. TEM foils were fabricated on a Tenupol-5 apparatus by double-sided jet electrolytic polishing. A solution of 5% perchloric acid (HClO_4_), 35% 1-butanol (CH_3_(CH_2_)_3_OH), and methanol as the electrolyte was used.

## 3. Results

To determine the degree of shear strain *γ_real_* achieved, and consequently the degree of slippage in metals during HPT, several experiments were performed using the “joint HPT of two halves” technique.

### 3.1. Experiment I

The initial copper disk was cut into two halves. Then these segments were subjected to joint HPT with the number of revolutions *n* = ¼ (anvil rotation angle *θ* = 90°) according to the scheme of Figure 1a. Based on the geometric characteristics of the obtained halves, it is possible to estimate the real degree of shear strain during HPT: *γ_real_* = *x/h*, where *x* is the displacement of the upper and lower surface of the halves, *h* is the thickness of the sample. From Figure 1b, it can be seen that the relative shift of the upper and lower surface of the Cu halves is close to the expected.

### 3.2. Experiment II

The one-piece Cu disk at the first stage was subjected to HPT *n* = 5, then the obtained HPT disk was cut into two halves (Figure 1c). The obtained halves were placed on anvils according to the scheme (Figure 1a) and subjected to joint HPT with the number of revolutions *n* = ¼. The change in appearance (Figure 1d) indicates some deformation, but the displacement of the upper and lower surfaces of the halves by shear is absent, and the picture differs significantly from that in Figure 1b. The degree of strain *γ_real_* in the experiment (II) is difficult to estimate, but the strain *γ_real_* is less than expected. Thus, after preliminary HPT c *n* = 5, slippages are observed even on relatively soft Cu.

### 3.3. Experiment III

Two halves of the Ti disk (Figure 2a) were subjected to joint HPT with the number of revolutions *n* = 1 on anvils with a groove diameter of 20 mm and a depth of 0.5 mm. After HPT, the appearance of the sample changed significantly, flash was formed, and the sample somewhat “upsetted” under the action of significant pressure. (Figure 2b). However, there was no significant shift of the upper and lower surfaces of the sample (Figure 2b). Hence, the real shear strain *γ_real_* of Ti after HPT *n* = 1 on grooved anvils is much smaller than *γ* = 7, which is predicted by Formula (1). This can be explained by the fact that noticeable slippage occurs already at the initial stage of HPT of Ti.

### 3.4. Experiment IV

The two halves of the Ti disk (Figure 2c) were subjected to joint HPT with *n* = 1/4 on flat anvils with a diameter of 20 mm. It can be seen from Figure 2c that the relative shear of the upper and lower surface of the Ti halves is quite large, and the degree of strain is close to that expected for HPT with *n* = 1/4.

### 3.5. Experiment V

The one-piece Ti disk was first subjected to HPT *n* = 5, then the obtained HPT disk was cut into two halves (Figure 2d). The obtained halves were placed on anvils according to the scheme (Figure 1a) and subjected to joint HPT *n* = 1 (Figure 2e). The appearance of the sample almost did not change (Figure 2e); the shift of the upper and lower surfaces of the halves on most parts of the sample is absent. The degree of strain *γ_real_* is less than expected. Thus, after the preliminary HPT with *n* = 5, a complete slippage is observed on the Ti sample.

Thus, the slippage effect is observed in the HPT of different metallic materials. The slippage effect was not so significant in the initial stages of HPT Cu. However, the slippage effect at HPT Cu becomes more significant after preliminary HPT with *n* = 5. Titanium received significant torsional deformation in the initial stages of HPT on flat anvils. For HPT Ti on grooved anvils, the slippage effect is already observed in the initial stages of HPT, and after preliminary HPT, for example with *n* = 5, the slippage effect becomes very significant, and no torsional deformation is realized in HPT.

Deformation at HPT occurs if friction force *Fμ* between the surface of the sample and the surface of the anvils is greater than the yield stress (YS) of the material:(2)Fμ>YS
(3)Fμ=P·μ

The friction force *Fμ* = *Pμ* (3), where *P* is the pressure, and *μ* is the friction coefficient. Anvils for HPT are mostly made of high-strength tool steel.

The coefficient of sliding friction *μ* in the copper-steel pair is 0.3. The yield strength (YS) of the original (undeformed copper) is about 300 MPa. YS of copper after HPT treatment is a maximum of 800 MPa [1]. If in Equation (2) *μ* = 0.3, hence the pressure for torsional deformation at HPT of initial copper (YS = 300 MPa) should be about 1 GPa, and the pressure for deformation of HPT-strengthened copper (YS = 800 MPa) should be about 2.7 GPa. It should be noted that in most HPT experiments [1,2,3], the authors indicate the applied pressure as 5 or 6 GPa. Thus, at these parameters, the slippage during the HPT of copper should not be significant.

The coefficient of sliding friction in the steel-steel pair is 0.15–0.2 [24]. The yield strength of the undeformed steel Fe-1%C is about 500 MPa. However, it is known that during deformation by the HPT method the steel at the early stages of HPT hardens and at HPT *n* > 1 its YS increases to 1500 MPa and more. As shown above, condition (2) must be fulfilled for torsional deformation by HPT. Taking *μ* = 0.2 we obtain that the pressure for torsional deformation under the HPT scheme of initial steel must be at least 2.5 GPa, and the pressure for deformation of steel hardened in the first stages of HPT must be at least 7.5 GPa. We should note that in most works [1,2,3] on HPT of steel, the applied pressure is 6 GPa or less. This explains the fact that in [22] a complete slippage at HPT of low-carbon steel Fe-1%C was observed as the degree of HPT increased. At the same time, it should be noted that the authors of this work observed the formation of a nanostructured state in the steel, similar to the HPT of steel Fe-1%C observed by other authors.

In the case of HPT Ti, the sliding friction coefficient in the steel-titanium pair, which is 0.4, must be used to analyze the slippage effect [24]. If the applied pressure is 6 GPa, then according to Formula (3), the friction force per unit area *Fμ* = 2.4 GPa. The YS of nanostructured titanium after HPT is about 2 GPa, respectively, with HPT Ti and a pressure of 6 GPa no slippage should be observed (*Fμ* > YS)

However, the pressure applied during HPT must be further analyzed. The pressure can be calculated as *P* = *U/S*, where *U* is the force and *S* is the area over which the force is applied. Researchers usually take the anvil area as *S* [3]. However, in reality, the area of the anvil–material contact should be larger than the area of only the anvils, taking into account the edges of the anvil working area and the material flash, which reduces the specific pressure [22]. In our case, the pressing force is *U* = 200 tonne, and the diameter of the anvil working part *d* = 0.02 m, hence, the anvil area $S=3.2\cdot 10^{-4}\;m^{2}$, and for this area, the pressure will be *P* = 6 GPa.

Nevertheless, taking into account the material tearing, the diameter of the sample after HPT is *d* = 0.025 m. Hence the sample area is $S\approx 5\cdot 10^{-4}\;m^{2}$. If we assume that the sample–anvil contact area is $S\approx 4\cdot 10^{-4}\;m^{2}$ and the actual pressure is much lower, about 5 GPa. Hence *Fμ ≈* 2 GPa, which is more than YS of original titanium (0.7 GPa), but less or comparable to YS of nanostructured titanium after HPT (up to 2 GPa). Accordingly, slippage is observed (*Fμ* ≤ *YS*), as shown by the above experiment V.

Previously, in numerous studies, the so-called pressure-dependent torsional moment estimation method was used to determine the presence of the slippage effect [16]. The torque sensor is mounted on the HPT installation, and in the case of HPT of any material, a curve for estimating the torque depending on the applied pressure is built [16]. According to this technique, the shear strength curve as a function of pressure, at pressures from 0 to a certain critical *P_k_*, follows one straight line due to the slippage of the anvil on the sample (the slope of the curve is determined by the true coefficient of friction of the sample material—anvil material and $S\;\sim \;\mu_{app}\cdot P$). At some *P_k_*, there is a “break” in the shear strength dependence—*P*. It is considered that such a break indicates that friction force in the anvil–sample contact becomes more than material yield strength (*Fμ* < *YS*), torsion of sample material begins, and the slope of the curve is determined not by the coefficient of friction of sample material—anvil material, but by an apparent “coefficient of friction” *μ_app_*, caused by the resistance of the flow of the sample [16].

However, the method of torque estimation depending on the pressure to determine the presence or absence of the slippage effect is certainly indirect and does not take into account several factors that may influence the presence of a break in the shear strength curve—*P*. Thus, in [25], it is shown that the coefficient of friction of metal–metal pair can decrease with increasing pressure. This should also change the course of the shear strength—*P* dependence and lead to a fold on the shear strength curve—*P* at a certain pressure. In addition, during HPT with an increase in the number of revolutions and a change in pressure, the area of the deformable sample changes in a complex way—the area increases as a result of sample upset, and at some stages of HPT it decreases due to the flow of flash from the contact zone. This changes the loading contact area during HPT and, accordingly, changes the course of the shear strength—*P* relation. Of course, the method of “joint torsion of disk halves” is a direct indication of the presence or absence of slippage during HPT.

The slippage effect depends in a complex way on several HPT parameters. These are such parameters as pressure, anvil design, anvil roughness, rotational speed, and others. It can be assumed that the slippage observed in our study is a particular case and is caused by the parameters/conditions of the HPT used in this work. In this regard, it is of interest to study the microstructure of titanium samples subjected to HPT on our equipment and in the HPT conditions commonly used by other authors. For this purpose, titanium disks were subjected to HPT with different rotational speeds at *P* = 6 GPa at room temperature.

After HPT by all conditions, a microhardness increase of more than two times is observed, which testifies to strong structure fragmentation (Table 2). HV values after HPT *n* = 5 are smaller in the center of the samples compared to the area in the middle of the radius (*1/2R*). This is a known result associated with a smaller degree of strain in the center of the sample [1]. As the number of revolutions increases up to *n* = 10, a noticeable increase in HV, compared to HV after HPT at *n* = 5, is practically not observed. The slowing of the growth of HV with an increase in the number of HPT revolutions over *n* > 5 is usually associated with the so-called “structural refinement limit”. It is assumed that there comes an equilibrium of the processes of accumulation of defects during deformation and their relaxation, and with further growth of the degree of deformation the structure is no longer refined, and HV does not increase. However, we can also assume that the slowing down of HV growth is due to slippage with an increase in *n* > 5. In addition, the microhardness does not increase with an increase in the number of revolutions, since the content of the hard ω-phase even slightly decreases in this case (see below).

X-ray diffraction analysis (Figure 3, Table 2) showed that, as a result of HPT with *n* = 5, microdistortions grow and the sizes of coherent scattering regions decrease compared to the initial state. As the number of HPT revolutions increases from *n* = 5 to *n* = 10, the microdistortions additionally grow slightly and the sizes of the coherent scattering regions decrease, indicating some additional refinement of the structure with increasing *n*, which, however, does not result in an appreciable increase in HV. The lattice microdistortions and the sizes of the Ti coherent scattering regions after HPT according to the selected conditions reach values close to similar parameters achieved by HPT CP Ti in the works of other authors [9,12,13].

In all conditions, the appearance of ω-phase due to α→ω transition is observed, as shown earlier [6,13]. After HPT *n* = 5, the ω-phase content reaches 50%. However, with a further increase in the number of revolutions *n*, the ω-phase content decreases to 20% at *n* = 10. A decrease in the ω-phase content with an increase in the number of revolutions above some critical number was noted earlier also by other authors [6,13]. The physical reason for a decrease in the ω-phase content with an increase in the *n* of HPT is not entirely clear, although in [6,13] this is explained by grain refinement below some critical size.

Analysis of the TEM studies’ results of Ti after HPT *n* = 10 shows: on the electronogram, along the ring, there are many blurred reflexes, (Figure 4c), which indicates a significant refinement of the structure. The light-field TEM image is difficult to analyze due to strong structure refinement, high dislocation density, and internal stresses. The dark-field image shows a refinement structure with a grain size of about 100 nm. The observed grain size is close to the grain size in HPT CP Ti noted in the works of other authors [9,12,13].

How can strain accumulate in the sample at HPT if slippage occurs? One of the possible explanations: it can be argued that the planes of the upper and lower anvils are inclined from each other by a small degree. Figure 5 shows a view of a Ti disk sample after HPT.

According to measurements, the sample has a non-uniform thickness at the edges (sample diameter 20 mm). At one edge, the thickness is 0.71 mm and at the other edge, it is 0.66 mm. This can be explained by the fact that the upper and lower anvils are tilted relative to each other at an angle of about 0.15°. In this case, when the anvils rotate relative to each other, the sample material under a pressure equal to several 6 GPa will flow from one zone of the deformation under the anvils to another. This will deform the material and modify its structure [16].

The case with such a deformation scheme (with an angle equal to 0.15°) was reproduced using finite-element computer modeling using the Deform 3D software package (Figure 6). As in the real experiment, a disk with a diameter of 20 mm made of technically pure titanium was taken as an initial workpiece. The thickness of the workpiece in the initial state was 0.6 mm. The strain-hardening curves are set based on literature data [24], the material is assumed to be plastic and isotropic, and there are no initial stresses and strains in it, and the anvils are set as a rigid body.

Three-dimensional solid models of the workpiece and anvils were created using the CAD system KOMPAS-3D and saved in “stl” format. A grid of finite elements—tetrahedrons—was generated. The number of finite elements was chosen based on preliminary calculations and was 75,000. The modeling was carried out taking into account the volume compensation of the workpiece model. The anvil models were not broken into a finite-element mesh.

Particular attention was paid to the contact conditions since the friction force between the anvils and the workpiece determines the principal possibility of implementing the HPT process. For a comparative analysis, two variants of the friction coefficient were investigated. Because the calculation of the volume deformation scheme with high contact stresses was carried out, the contact conditions were set using the Siebel friction factor. The friction factors between the workpiece and the rotating anvils were taken to be 0.05 and 0.7, respectively. The contact between the stationary anvil and the workpiece was set using the sticking function. On the contact surfaces of each anvil, the impermeability condition was set.

Modeling was divided into 2 stages, at the first of which the sample was upset at the value of about 0.1 mm to fill the cavity of the groove of the lower anvil and to ensure full contact of the workpiece with the upper (rotating) anvil. Then the torsion operation was simulated, and the anvil rotation speed was chosen constant and equal to 1 rpm.

The simulation was performed with a constant time step of 0.1 s. The method of conjugate gradients was used. The finite-element model described the motion of a continuous medium based on the Lagrangian approach.

Figure 7 shows the distribution of strain intensity (e_i_) after 1 and 5 revolutions of the HPT.

The analysis of the obtained data shows that even for the case *μ* = 0.05 (which presupposes that the anvil slips over the workpiece surface) at HPT *n* = 1 the value of the accumulated strain intensity varies from *e ≈* 0.6 to *e ≈* 5 at the edges of the sample. The strain has an uneven distribution over the sample—more at the edges of the sample, less in the center. However, a significant area of the sample received strain from 0.875 to 1.75. At HPT *n = 5* revolutions, the accumulated strain intensity further increased slightly, as the area of the region with strain from 0.875 to 1.75 increased markedly. It shows that the scheme in which “upper and lower anvils are inclined relative to each other at an angle of about 0.15°” leads to accumulation of strain with the growth of the number of revolutions from *n* = 1 to *n* = 5. It should be noted that the value *e ≈* 1.75 leads to a significant refinement of the structure and, taking into account the high applied pressure, formation of nanoscale structural elements is possible.

## 4. Conclusions

T slippage effect during high pressure torsion deformation of Cu and Ti samples was investigated. The “joint torsion of the disk halves” method was used to evaluate the slippage effect. It was shown that for relatively “soft” materials such as Cu, the slippage during HPT in the initial stages (*n* = 1/4) is negligible. In the case of harder and tougher materials, e.g., Ti, slippage already starts in the initial stages of HPT. After high pressure torsion with a revolution number of *n* = 5 with a further increase in *n* the slippage is complete and no torsional deformation occurs. The slippage effect is explained by insufficient friction forces in the sample surface–anvil surface pair.

HV, TEM, and XRD studies have shown that the structure of Ti after HPT samples according to the used conditions is refinement to a nanocrystalline state. A structure similar to that observed in Ti after HPT by other authors is formed. The dislocation density increases with increasing HPT degree from *n* = 5 to *n* = 10 revolutions, despite slippage. A model has been proposed to explain the accumulation of strain in the sample at HPT with *n* > 5. According to this model, it is assumed that the planes of the upper and lower anvil at HPT are slightly inclined relative to each other. Mathematical finite element simulations have shown that this leads to an accumulation of significant strain in the sample as the anvils rotate, even taking into account the slip effect inherent in the contact conditions.

## Figures and Tables

**Figure 1 materials-16-00162-f001:**
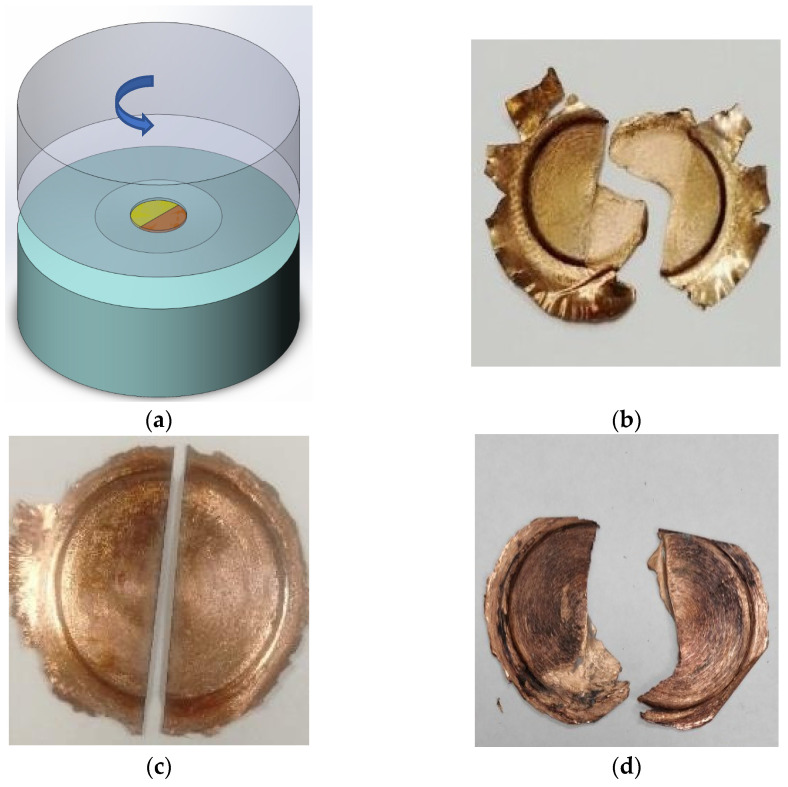
Schematic of joint HPT halves to determine the degree of shear strain (**a**), copper halves disc after joint HPT *n* = 1/4 (**b**), copper disk after HPT *n* = 5 cut into halves (**c**), same halves after subsequent joint HPT *n* = 1/4 (**d**).

**Figure 2 materials-16-00162-f002:**
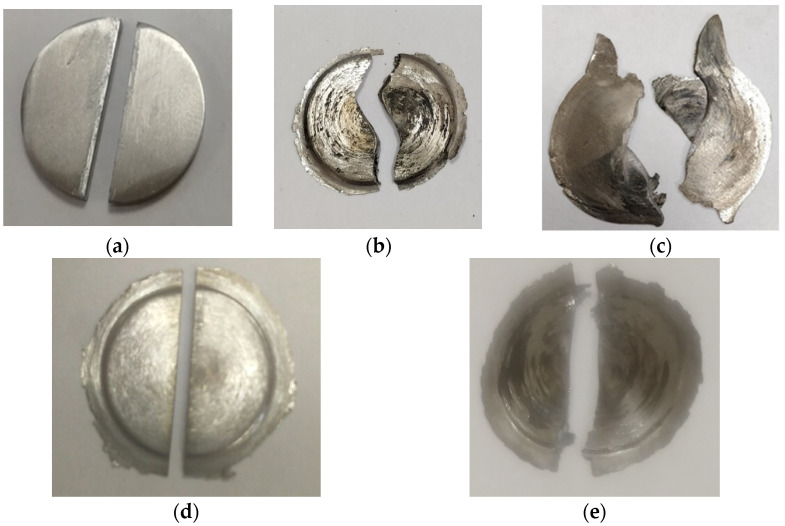
Two Ti disk halves d = 20 mm prepared for joint HPT (**a**), (**b**) Ti halves after joint HPT *n* = 1 on grooved anvils, (**c**) Ti halves after joint HPT *n* = ¼ on flat anvils; (**d**) Ti disk after HPT *n* = 5 cut for joint HPT; (**e**) the same halves after subsequent joint HPT *n* = 1 (*n* = 5 + 1).

**Figure 3 materials-16-00162-f003:**
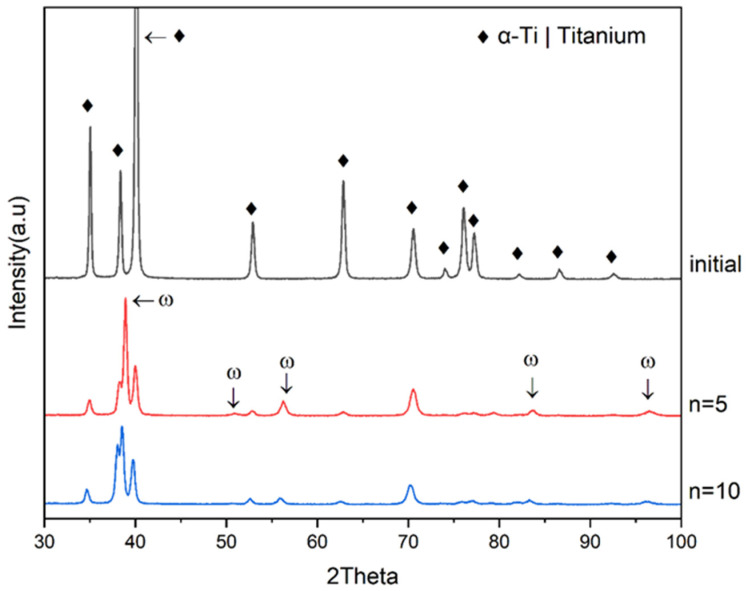
XRD CP-Ti in an initial state and after HPT *n* = 5 and HPT *n* = 10.

**Figure 4 materials-16-00162-f004:**
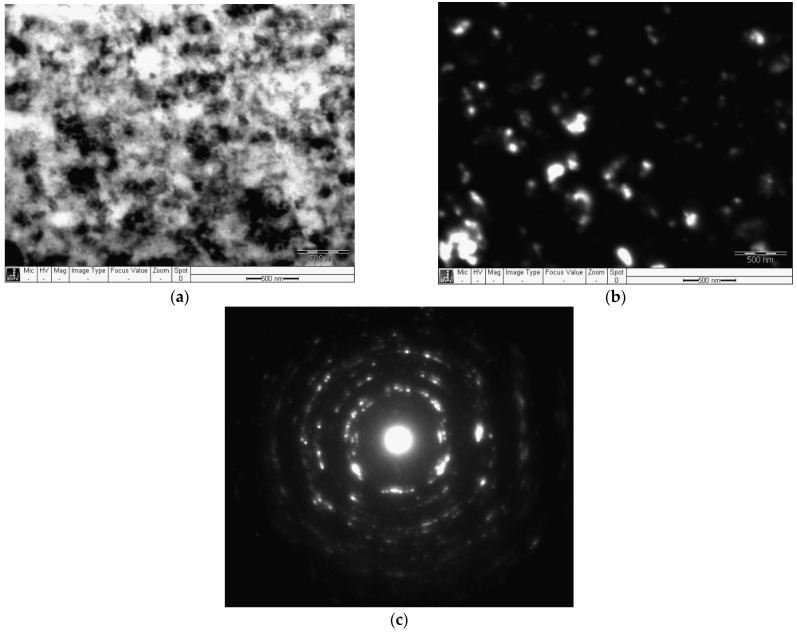
Structure of Ti after HPT *n* = 10: (**a**) bright field; (**b**) dark field; (**c**) electronogram.

**Figure 5 materials-16-00162-f005:**
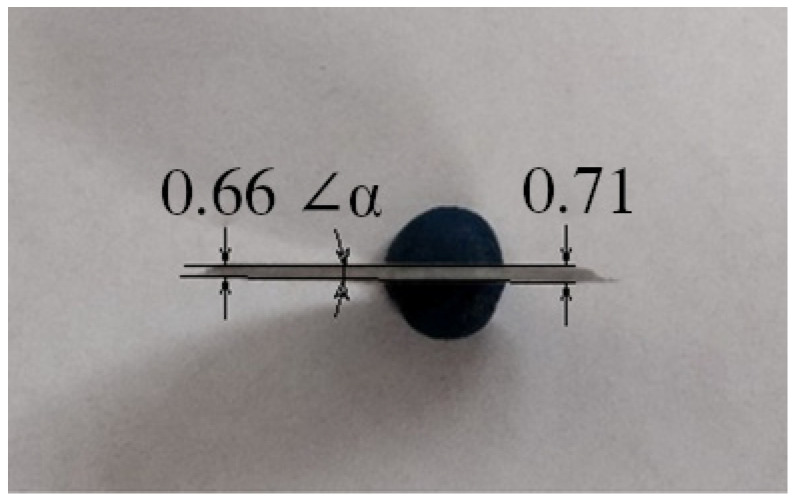
Cross-section of the sample after HPT with a height difference forming an angle = 0.15 degrees.

**Figure 6 materials-16-00162-f006:**
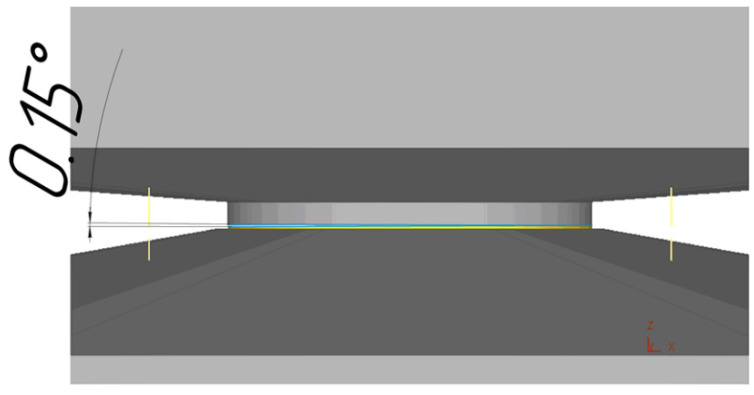
Scheme of the HPT in which the anvils are at an angle of 0.15 ° to each other.

**Figure 7 materials-16-00162-f007:**
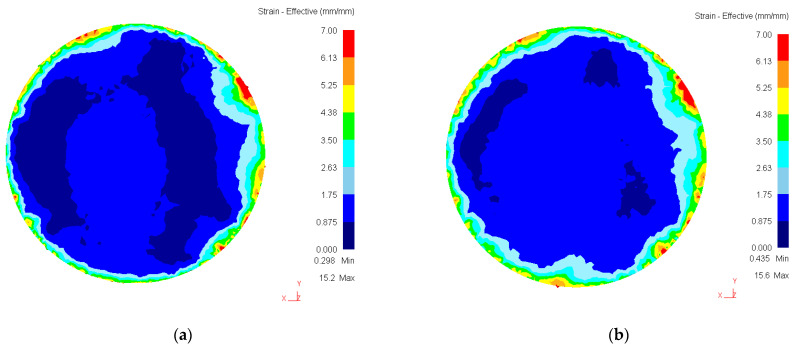
Distribution of accumulated strain intensity after 1 and 5 revolutions of HPT: (**a**) 1 revolution; (**b**) 5 revolutions.

**Table 1 materials-16-00162-t001:** Chemical composition of titanium VT1-0.

Ti	Al	Si	Fe	C	O	N	H	Remaining Impurities
Base	0.010	0.002	0.120	0.004	0.143	0.003	0.0008	0.077

**Table 2 materials-16-00162-t002:** Results of HV measurements and the XRD analysis of Ti before and after HPT.

Number of Revolutions, N	Measurement Region	HV0.1	Lattice Distortion, %	CSR, * nm	ω-Phase Fraction, %
Initial	-	210	0.17	72	0
5	Center	455	0.25	23	46
	1/2 R	475			
10	Center	440	0.34	16	20
	1/2 R	450			

* CSR—Coherent scattering region.

## Data Availability

All data generated or analysed during this study are included in the published article, and are available from the corresponding authors upon request.

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
