# Peer review of "Some Slippage Issues in High-Pressure Torsion Using Cu and Ti Samples as an Example"

_materials, 2022, doi:10.3390/ma16010162_

Round 1
Reviewer 1 Report
The effect of slippage during High Pressure Torsion (HPT) of technically pure Ti samples was investigated in this work. The "joint torsion of the disk halves" method was used to evaluate the effect of slippage. This study is of great significance to the study of plastic deformation mechanism of metal materials. The research is interesting, but there are still some questions.
1. The title of this work is “Some slippage issues in high-pressure torsion using Ti samples as an example”, This paper takes titanium as the research object to analyze the slippage issues. However, the slippage behavior of different crystal systems and different alloys is not the same, so the title of the paper is suggested to be revised more precisely.
2. In order to study slippage problem in this paper, high resolution analysis is very important. But the results given in this paper are not ideal. I suggest the authors provide clearer high resolution and diffraction patterns. It should be analyzed from the angle of crystal plane spacing and dislocation.
3. Line 189, Figure 5d should be revised to Figure 2d?
4. Line 421-423,Fe-0.1%C is mentioned in the conclusion, but there is no relevant data of this alloy in the paper. Please explain how you reached your conclusion.
5. In Figure 5, 0,66 should be changed to 0.66, and 0,71 should be changed to 0.71.
6. Line 18-19: A model is proposed to explain the accumulated strain in the sample at HPT.
What exactly does this model entail? What are the experimental results of the support? If authors mention this model, they need to provide sufficient evidence.
Author Response
The title of this work is “Some slippage issues in high-pressure torsion using Ti samples as an example”, This paper takes titanium as the research object to analyze the slippage issues. However, the slippage behavior of different crystal systems and different alloys is not the same, so the title of the paper is suggested to be revised more precisely.
Thank you! Changed the title to “Some slippage issues in high-pressure torsion using Cu and Ti samples as an example.”
- In order to study slippage problem in this paper, high resolution analysis is very important. But the results given in this paper are not ideal. I suggest the authors provide clearer high resolution and diffraction patterns. It should be analyzed from the angle of crystal plane spacing and dislocation.
Thanks for the comments. We agree, high resolution analysis is important. However, the structure after HPT is very fine, and the quality of the TEM we have is insufficient for high resolution analysis and detection of dislocations. The increase in dislocation density is indirectly evidenced by the XRD data, the “Lattice distortion” parameter, Table 2. But the main goal of the TEM and XRD studies presented in the article is as follows: looking at slippage) is close to the structure in HPT CP Ti observed in the works of other authors [9,12,13]. Thus, 1) slippage at HPT Ti was also possible in other authors, although they did not track slippage 2) strain accumulate in the sample at HPT, and a nanostructural state is formed if slippage occurs
- Line 189, Figure 5d should be revised to Figure 2d?
Thanks, the correction has been made, Figure 5d revised to Figure 2d.
- Line 421-423,Fe-0.1%C is mentioned in the conclusion, but there is no relevant data of this alloy in the paper. Please explain how you reached your conclusion.
Thanks! this paragraph has been removed.
- In Figure 5, 0,66 should be changed to 0.66, and 0,71 should be changed to 0.71.
Thanks! corrected
- Line 18-19: «A model is proposed to explain the accumulated strain in the sample at HPT.»
What exactly does this model entail? What are the experimental results of the support? If authors mention this model, they need to provide sufficient evidence.
Thanks, Change made:
The following assumptions are made to explain the accumulated strain in the sample at HPT. It is assumed that the planes of the upper and lower anvil during HPT are at a slight in-clination relative to each other. Computer modeling using the Deform 3D software package has shown that this leads to the accumulations of significant strain during HPT.
Reviewer 2 Report
Dear Authors,
Please find below my comments/observations regarding your manuscript:
1. Considering that the experiments included also sample of Cu alongside with Ti (even as auxiliar material), the title should include “Cu” also: Some slippage issues in high-pressure torsion using Cu and Ti samples as an example. Please add it.
2. In addition, you have included in several places a discussion that refers to Fe-0.1%C, for comparison, even for the conclusions part. It would be good, in my opinion, to add an explanation of this approach; otherwise, it seems a bit confusing: the title talks only about titanium, and in the text, you have titanium, copper and steel. Therefore, it would be good to have a clear explanation of what you had in mind.
3. It should be explained what means CSR in the table 2. Also, please add the SI units for each parameter indicated in table 2.
4. Also referring to Table 2, please add a possible explanation for the halving of omega percentage from n-5 to n-10 (from 46% to 20%). Why does this happen? Please add a presumed explanation for the structural behaviour/transformation.
5. Also, how do you explain the fact that the hardness HV (table 2) remains practically unchanged even if the amount of omega is halved? Only by grain refinement? And what is the percentage of the grain refinement from n-5 to n-10? Here, the analysis should be extended with more details.
Author Response
- Considering that the experiments included also sample of Cu alongside with Ti (even as auxiliar material), the title should include “Cu” also: Some slippage issues in high-pressure torsion using Cu and Ti samples as an example. Please add it.
Thank your for comments ! The title has been changed
- In addition, you have included in several places a discussion that refers to Fe-0.1%C, for comparison, even for the conclusions part. It would be good, in my opinion, to add an explanation of this approach; otherwise, it seems a bit confusing: the title talks only about titanium, and in the text, you have titanium, copper and steel. Therefore, it would be good to have a clear explanation of what you had in mind.
Thank you for your comment. this paragraph of Fe-0.1%C removed
- It should be explained what means CSR in the table 2. Also, please add the SI units for each parameter indicated in table 2.
Thank you for your comment. Thanks, the fix has been made. CSR - coherent scattering region determined from XRD
- Also referring to Table 2, please add a possible explanation for the halving of omega percentage from n-5 to n-10 (from 46% to 20%). Why does this happen? Please add a presumed explanation for the structural behaviour/transformation.
Thank you for your comment. The following corrections have been introduced into the article (line 289): However, with a further increase in the number of revolutions n, the ω-phase content decreases to 20% at n=10. A decrease of ω-phase content with an increase in the number of revolutions above some critical number was noted earlier also by other authors [6,13]. The physical reason for a decrease of ω-phase content with an increase in the n of HPT is not entirely clear, although in [6,13] this is explained by grain refinement below some critical size.
- Also, how do you explain the fact that the hardness HV (table 2) remains practically unchanged even if the amount of omega is halved? Only by grain refinement? And what is the percentage of the grain refinement from n-5 to n-10? Here, the analysis should be extended with more details.
Thank you for your comment. The following corrections have been introduced into the article: In addition, HV does not increase and even decreases within the measurement error with increasing n HPT from n= 5 to n=10 , since the content of the solid ω-phase decreases slightly with increasing n HPT (see below)
Round 2
Reviewer 1 Report
The author made corresponding modifications according to the suggestions of the last review, and most of the modifications met the requirements. However, there is no high-resolution analysis of the tissue, which is attributed to the fact that the tissue is too fine. I think this can be achieved by equipment with higher functions. But the current status can also be published. In conclusion, I agree to publish the paper.